# Pathways through homelessness among women in Addis Ababa, Ethiopia: A qualitative study

**Kibrom Haile**[1]*, **Halima Umer**[2], **Tolesa Fanta**[1], **Addis Birhanu**[1], **Edao Fejo**[1], **Yirga Tilahun**[2], **Habtamu Derajew**[1], **Agitu Tadesse**[1], **Gebreselassie Zienawi**[2], **Asrat Chaka**[1], **Woynabeba Damene**[1]

1 Research and Training Department, St Amanuel Mental Specialized Hospital, Addis Ababa, Ethiopia,
2 Clinical Department, St Amanuel Mental Specialized Hospital, Addis Ababa, Ethiopia

* kibromhaile10@yahoo.com

**Data Availability Statement:** The datasets generated during and/or analyzed during the current study are not publicly available due to

## Abstract

### Background

There has been a paradigm shift in understanding homelessness. The shift is from the belief that homelessness results from lack of secure housing towards the view which explains homelessness in terms of the complex interactions of factors which determine the pathways into and out of homelessness. The evidence base for women's homelessness is less robust than men's homelessness. The effect of gender and its relationship with homelessness has been neglected. Addis Ababa, the capital city of Ethiopia, is estimated to be home for around 50,000 homeless people. This study aims to explore pathways through homelessness in women who were sheltered in a facility for the homeless in Addis Ababa.

### Methods

In-depth interviews were conducted in 2019 with 14 women who were 'roofless', and were gathered for support in a temporary shelter in Addis Ababa, Ethiopia. The shelter was one of the eight such facilities established in Addis Ababa few months earlier than the study. For data analysis the QDA Miner 5.0.30 software was used and data was analyzed using the-matic analysis approach.

### Results

The analysis revealed that determinant factors for pathways into homelessness among women occurred on the background of predisposing factors, such as poverty, being raised by caregivers other than biological parents, child marriage, unstable employment history. On top of the predisposing factors listed above the occurrence of precipitating factors such as problems with marriage, migration, death of parents, deception, became the immediate cause of homelessness. Despite mentions of positive experiences of homelessness such as mutual support and good social life within network of homeless people, the net effect of the interaction between negative and positive experiences of the homeless life, together with the effectiveness of coping strategies by the participants resulted in the participants'

ethical restrictions and for protection of participant privacy. Because this is a qualitative study and the data includes sensitive life stories of the participants which, if publicly available, could result in the possibility of damage to the participants. The authors believe that even if anonymous, there is a possibility of the participants being identified from the other personal information included in the data which cannot be totally excluded lest could remove important information used during data analysis. The authors believe that they have the duty to protect the privacy and confidentiality of their study participants as required by all research ethics standards. The participants have been assured about data privacy and confidentiality before they gave consent to participate. Therefore, the reason for not publicly sharing the data is authors' discretion and duty to protect participant privacy. However, the Institutional Review Board (IRB) of St Amanuel Mental Specialized Hospital also has the responsibility to monitor and follow the ethical undertaking of matters concerning the study. Contact information for the IRB is as follows: amsh_res@amsh.gov.et.

**Funding:** The author(s) received no specific funding for this work.

**Competing interests:** The authors have declared that no competing interests exist.

**Abbreviations:** AIDS, Acquired Immune Deficiency Syndrome; ETB, Ethiopian Birr; FEANTSA, European Federation of National Organizations working with the Homeless; HIV, Human Immune-deficiency Virus; GDP, Gross Domestic Product.

decision of whether homelessness is tolerable. Finally, the presence of perpetuating factors such as lack of affordable house, feeling of shame to go back home, and unfavorable situation at home discouraged participants from exiting the homeless situation.

## Conclusion

From the findings of the study we conclude that the predisposing factors and the precipitating factors resulted in the occurrence of onset of homelessness among the participants. Once homeless, the experiences of life as homeless, and the availability of the means to exit from it determined whether the participants would stay homeless or exit from it.

## Background

### Understanding homelessness

The term 'homelessness' does not give identical meaning across nations or across the research community [1]. Nowadays, it is becoming evident that a universal definition of homelessness is neither useful nor necessary. The understanding of homelessness may differ across high-income and low-income country contexts. Therefore, a universal standard definition would not serve the development of context-specific intervention and policy [2,3]. The definition of homelessness can be made to be wider to include bigger population groups, or narrower to prioritize scarce resources to the most needy. However, any definition of 'homelessness' must be based on certain key principles. One such principle assumes that homelessness should not be taken only as the absence of a permanent accommodation [1]. According to this principle the definition of homelessness includes the existence of deprivations across a number of dimensions. These dimensions may be physiological (such as lack of bodily comfort or warmth), emotional (such as lack of love or joy), territorial (such as lack of privacy), ontological (such as lack of rootedness in the world), or spiritual (such as lack of hope) [4]. For instance, for a woman who leaves house due to abuse, it seems that "housing is the problem for which homelessness could be the solution" [5].

Several authors give emphasis to multidimensional understanding of homelessness [6–10]. Accordingly, homelessness can be best understood in terms of the pathways which lead into and out of the situation [1,4]. This kind of understanding provides a complete picture of homelessness and offers the opportunity for better intervention measures. Homelessness pathway has been defined as 'the route of an individual or household into homelessness, their experience of homelessness, and their route out of homelessness into secure housing'.[11].

The other principle that must be used in defining homelessness is the specific communities' minimum standards for adequate accommodation [12]. The minimum standards for adequate accommodation may be different across countries and socio-economic situations [12]. The definition of homelessness developed by the European Federation of National Organizations working with the Homeless (FEANTSA) emphasized the importance of this principle. According to FEANTSA there are four different levels of homelessness including rooflessness (sleeping rough), houselessness (living in institutions or short-term 'guest' accommodation), insecure accommodation, and inferior (substandard) housing [2,3]. 'Hidden homelessness' is a good example for homelessness by way of 'houselessness' or 'insecure accommodation.' The hidden homeless population of women comprises those who double up with friends or family, or those staying at rooming houses or hotels [13]. Studies conducted in Ethiopia used the

word 'homelessness' to refer to those who were 'roofless' or 'street homeless' [14–16]. A previous study conducted in Addis Ababa used the definition of homeless as 'those sleeping in designated shelters or public spaces'.[17].

## Gender and homelessness

Women's homelessness is associated with domestic violence at much higher rates than is the case for men [18]. In fact, gender-based violence was a strong feature of women's homelessness [19]. The ugly truth, however, is that women may become homeless to escape domestic violence but homelessness is not likely to solve the problem. The truth is homeless women are more likely than homeless men to perceive and endure greater safety issues, including sexual victimization [20]. The other peculiar feature of women's homelessness is that they are more likely to be in the situation of 'hidden homelessness.' Studies show that the majority of women and girls experiencing homelessness are members of the 'hidden homeless', concealing the alarming rate of women's homelessness [19,21]. The occurrence of 'hidden homelessness' reveals that women tend to draw on informal resources to manage homelessness. Women are more likely than men to stay with friends, family and acquaintances at higher rates and probably for longer periods than men [18]. Because of their role in the family, women's homelessness is traditionally categorized as either 'family homelessness' or 'single homelessness' [22]. It must be noted that, unlike the case of men, women's homelessness is likely to take the form of 'family homelessness'. In this regard, homeless mothers unaccompanied by their children are likely to be labeled as 'undeserving' or 'bad' mothers [22].

## Causes of homelessness in women

Understanding the causes of homelessness in women is critical for implementing effective prevention measures. Delivery of need-based services to homeless women, and provision of effective support to help them achieve housing stability can be better accomplished if there is evidence about the cause of homelessness [23]. It must be noted, however, that causes of women's homelessness are wider and more systemic than the individualized concept of reasons would lead us to believe [24]. A majority of women experienced multiple adversities and deprivations during childhood, including poverty, neglect, housing instability, difficult family situations, as well as discontinuity in their schooling [19]. Studies identified causes such as domestic violence, poverty, violence in the community, history of early-onset use of substances and drugs, loss of employment, loss of access to affordable housing or losing house through eviction, mental illness, and history of childhood violence [23,25–27]. As high as 72% of homeless women had experienced violence and/or abuse during childhood, and two-third of them had experienced intimate partner violence [19]. Lack of affordable housing, poverty, exposure to violence and unsafe spaces are the most common reasons for women's homelessness [28]. According to studies conducted in big cities in Ethiopia, the factors associated with state of women's homelessness were migration, urbanization, poverty, abuse, escape from child marriage, unemployment, death of one or both parents, divorce of parents, pregnancy out of wedlock, poor educational status, and abandonment by spouse [14,15,29,30].

## Challenges of homelessness in women

The challenges of homeless women extend beyond simply finding shelter [25]. They also are troubled with finding work and they suffered from feelings of sadness, vulnerability and stress more than homeless men [25]. Once homeless, hardship and challenges are daily realities for women. Sexual assault, mental illness and substance abuse were identified to be high [23,31]. Women reported being stigmatized, verbally and sexually assaulted and higher risks of

violence and sexual abuse while sleeping rough [18]. Homeless women carry multiple stigmas and labels such as 'bad mother', 'prostitute', etc which makes it difficult for them to seek help; the stigma and labels also become a barrier for recovery from homelessness [32]. The conditions of sleeping rough meant that women concealed themselves or kept moving at night, including the need to conceal their gender by dressing as men [18]. Homeless women who worked as prostitute were exposed to intensive daily substance use, and to serving clients in public space [31]. In Ethiopia, homeless women were vulnerable to rape, physical abuse, emotional abuse, extreme poverty, hunger, physical safety concerns, lack of safe drinking water, poor sanitation and prevailing diseases [14,15,29,30]. Exposure to violence continues to be a common experience during homelessness, with 57% of women reported being a victim of violence [28].

## The need to conduct the current study

Worldwide, the evidence base is less robust about women's homelessness than men's homelessness [33–36]. Reports show that women are ignored in homelessness services due to the reason that homelessness is regarded as a problem of men [32]. In Ethiopia, likewise, the issue of female homelessness has been a neglected topic. The number of homeless girls and women in Ethiopia is growing [14]. In fact, from previous studies conducted in urban areas of Ethiopia, adult women accounted for one-third of the homeless population [37]. Despite the observable problem of homelessness among women in developing countries, there has been lack of studies conducted to determine pathways through homelessness in such countries. This qualitative explorative study was aimed to address this important issue among women in Addis Ababa, Ethiopia. For the purpose of our study 'homeless' referred to those who slept in designated public spaces or shelters, or those who were 'roofless.'

## Method

A qualitative study was conducted in 2019 to explore homelessness pathways among female residents of a shelter facility in Addis Ababa. The second and sixth authors were volunteer workers of the shelter and they had established prior relationship with the participants; however, whenever those volunteer workers were involved in the research activities they said so to the participants and assured them that their participation in the research activities was fully voluntary, and without any negative or positive consequences upon the services they received in the shelter facility.

## Setting

The study was conducted in Addis Ababa, which is the capital and largest city in Ethiopia. The current population size of Addis Ababa is estimated at 5–7 million. The city is the economic powerhouse of the country, with a major share of the country's GDP generated from it. About 40% of Addis Ababa's workforce is government employees, 31% private organization employees, 25% own-account workers, 2% employers, and another 2% unpaid family workers and others. Females account for 54.2% of the population of Addis Ababa, and one-quarter of the population is below the age of 15 years. One in four of the women in the city have no education [38]. According to the city administration estimates, there are around 50,000 homeless individuals in the city [16].

The participants of the study were recruited from a temporary shelter which was constructed in Addis Ababa about 4 months prior to the beginning of the study. The shelter was established to provide services to homeless adult women with their dependent children if they had any, and elderly men. The shelter was providing services for up to 250 people at the time

of the study. Recruitment of the women to the shelter was from all sub-cities of Addis Ababa and on voluntary basis. The shelter was established and funded by the government and was run in collaboration with volunteer workers. Food, clothes and shelter was provided by the facility, as well as medical, mental and psychosocial services. The beneficiaries stayed at the shelter for maximum of 5 months including the 3 months allocated for vocational training which included hair-dressing, cleaning and tailoring.

## Sampling strategy

A purposive sampling technique of typical case sampling was employed to include adult women participants who could provide rich data about their homeless life experience. A face-to-face in-depth interviewing technique was conducted to collect data from participants. Data analysis was made based on the life experiences of respondents, as well as their feelings, perceptions, understanding and expectations about their experiences. The meaning they gave to each of their experiences was also used for data analysis.

## Data collection procedure

One-on-one in-depth interviews were conducted to collect data from participants until a point of theoretical saturation was achieved. Interview was conducted by two interviewers, who are the first two authors. One of the interviewers was a practicing psychiatrist; the other interviewer was a practicing mental health professional who had MSc degree in the profession of mental health. Both interviewers were practicing clinicians and researchers who were working at the only mental health hospital in Ethiopia. A total of 15 women were interviewed, but one of the interview data was not included in the analysis due to data quality issues. Interview was conducted by using open-ended interview guide questions. The average duration of the interviews was 25 minutes. All interviews were audio-recorded by using digital audio recorder application on a smart phone.

## Data analysis

The data was analyzed using the qualitative data analysis software QDA Miner 5.0.30. The recorded audio was first transcribed into text written in the local language Amharic. The text was read line-by-line and then translated into the English language. The data for each respondent were made into a word document; documents for all respondents were imported into the software as 'cases'. Some of the topics were identified in advance and included in the interview guide questions for further exploration, while others were derived from the data. Thematic analysis approach was used for analyzing data. A careful line-by-line reading of each document was done to identify major topics within the data; the topics were then labeled and entered into the 'variables' section of the software. The responses were coded and included under each relevant topic or 'variable' for further analysis.

During data analysis, themes were identified and those themes were grouped under the appropriate topic or 'variable'; similar topics or themes were merged, and some new topics or themes were also created during data analysis as new themes emerged. Some of the themes overlapped for some participants; however, the factors were counted separately to make a complete list of possibilities.

## Ethics approval and consent to participate

Ethical clearance was obtained from the Institutional Review Board of St Amanuel Mental Specialized Hospital. The participant information sheet was read to each participant before they

could give consent to participate. Permission was obtained from each participant to audio-record the interview. The interview was conducted after the participants gave signed consent to be interviewed. Interviews were conducted in private in an office room which was arranged for the purpose of the interview.

## Results

Participants were adult women whose age range is from 18 to 37 years. The participants' estimated total length of life as homeless varied from the shortest 1 year to the longest 19 years. One of the participants was born on the streets and had been homeless for all of the 19 years of her life. Four of the participants had no children, one of the participants was pregnant at the time of interview, and 9 of the participants had at least one child. Among those who had at least one child, two participants had 2 children, one participant had 3 children and the rest had 1 child each. The results of the qualitative data analysis are described below:

### Predisposing factors to homelessness

Predisposing factors indicate how the respondent lived since childhood before she became homeless. Predisposition indicated the level of vulnerability of the respondent to becoming homeless upon the occurrence of contextual events. In the study, poverty was identified as a remarkable predisposing factor for homelessness. Some participants described how they lived under poverty before they became homeless as follows:

> "………. my father's land was confiscated and he started tilling rented land, which gradually became hard for him as his physical capacity deteriorated. I then started to feel the hardship of life due to economic scarcity; this made me collect and sell grass to support myself."[11th participant]

> "I lived with my both parents at the beginning; one time my father abandoned us and left to another place. Then there was no one to cultivate the land for us and we became poor. At the time when we hardly had food to eat and got starved …….."[8th participant]

Being raised by people other than biological parents was a factor reported by some participants. Such participants had been raised by either their siblings, or uncles or aunts, or other relatives. One of the participants said:

> "I was born in Jimma Town. ………….. my parents died; they died when I was so little I don't remember when they died. I was raised by my brothers…..."[13th participant]

Related to the fact mentioned above, others were raised in a family where one of the parents was a step-parent. Some of the participants said they were uncomfortable with the step-parent's behavior, while others said they had no problem with it. However, they had perceived some form of discrimination or isolation, or another form of harsh treatment by the step-parent. Participants said:

> "I was a child when many relevant things happened; what I do remember is that I was raised by a stepmother till I was 7 years old. I don't know my mother. When my stepmother's son was going to school, I was not."[1st participant]

*"I was born in the rural part of Gojjam. I don't know my mother; she died when I was a small baby. I am the 5th child to my family. My father was raising me after my mother's death. I used to herd cattle and chased birds from the sorghum farm. With the push from his relatives, my father got married and I had a stepmother; but I wasn't comfortable with my step-mother."[9th participant]*

Migration to urban areas was another predisposing factor. One participant described how she moved from rural to urban area with expectation of better life; however, what awaited her was not what she expected. She said:

*"My sister, who lived in Addis, was not in favor of my marriage and she brought me to Addis Ababa with her. I expected to have a reasonably good life in my sister's home; she also had given me promise that I would attend school."[9th participant]*

The other predisposing factor was child marriage. In some parts of Ethiopia, particularly in the rural areas of the north-west, there is this culture of marriage at the earliest possible age. It is called 'madego' marriage, meaning the child or girl gets married at early age and will grow up under the custody of the in-laws till she becomes 'fit' for sexual or marital responsibilities. Being subjected to 'madego' was identified among the participants prior to their becoming homeless. One of the participants described her situation as follows:

*"I was born in the rural part of Gojjam; . . . . . . . . . . . I was forced into the marriage arrange-ment called 'madego' at age of 5; but I was uncomfortable with the marriage. It was uncom-fortable for me because, due to issues with dowry, both families were in dispute and I was in the middle of this dispute. I suffered from hatred, verbal and emotional abuse from both families."[12th participant]*

Some participants had experienced exploitative work conditions. Participants who migrated from outside of the city may not have had relatives inside of the city that they could rely on for support at times of difficulty. Such participants were likely to tolerate exploitation for sometime due to lack of options, but their situation makes them vulnerable to subsequent homelessness. One participant reported:

*". . ... One rich woman found me and took me with her. I started working in her hotel as cleaner; I do household chores in the evening. The work I started was so hectic I hardly had time to keep my personal hygiene; I was infested with lice. On top of this, I didn't have formal salary. People were advising me to quit."[13th participant]*

Similarly, other participants reported unstable work history. Those respondents had had frequent change of working environment. One participant said:

*"Especially, there was a demented old woman who is the mother of my madam. It was difficult for me to tolerate the behavior of the old woman and, after 3 months . . . . . . . I finally managed to find another household to work as housemaid; I worked for 7 months in that house and asked to visit my family back in Awaro."[11th participant]*

Commercial sex work was another factor which probably predisposed participants to homelessness. A participant described the following:

*"…….....My sister brought me to Addis to live with her; I was 10 years old that time. Later on I realized she was working as a commercial sex worker; I had no choice but to work the same as hers and I became a commercial sex worker."[11th participant]*

Some participants were raised up in a family in which there was domestic violence. One participant said:

*"My father used to come home drunk and hit my mother; then she left us and migrated to Addis Ababa."[7th participant]*

The other factors identified as possible predisposing factors were HIV infection, non-supportive family, and difficulties and problems related to marriage.

## Immediate reasons for becoming homeless

The 'immediate reasons' for becoming homeless topic refers to the triggering events which finally forced the participants into homelessness. The thematic areas which emerged under this topic are discussed below.

Problematic marriage was identified as immediate cause for homelessness. Such marital problems include a spouse who was involved in an extramarital affair or a spouse who was abusive. Participants said:

*"While I was living with my husband and making a family, he loved another woman. Then I left him and started living on the streets."[3rd participant]*

*"……….. But the man who made me pregnant was alcoholic and had a wife; he was abusive and he assaulted me on daily basis. I tolerated till my youngest daughter was one year and 8 months old; but one day I got desperate, took my child and left him. Then I started living on the street."[7th participant]*

Hard and intolerable work conditions precipitated participants' becoming homeless. One respondent said:

*"…Despite all the hardship, I worked for 5 years. Finally, I lost all my hope in life. I became so angry and frustrated I finally abandoned everything and went out to the streets to live there."[13th participant]*

Peer pressure was another immediate cause for becoming homeless. One participant described her situation the following way:

*"When I told my friends about how my sister punished me, they encouraged me to leave her house. They told me they had house; and I told them I would contribute some money to pay the house rent; we agreed with this idea. That day I didn't go home to my sister; instead, I went with my friends to their place. At their place, I didn't get what I expected; what they told me their 'home' was a place under an old bridge. They actually were living on the street. I say the main reason I ended up on streets was a pressure from a friend. Even if I was angry at my sister, had it not been for the pressure from my friend I wouldn't be on the streets." [9th participant]*

Related to the peer pressure described above, some participants were actually deceived into street life. One of them explained her experience as follows:

*"One day there was a girl of approximately my age. . . . . . . . . .. She intentionally made acquaintance with me and asked me to go to her home and play with her. We went together to Debre Zeit. While we were at Debre Zeit, she stole money from local bajaj taxis inside the station. It was my first time to see such way of theft. Then she told me we should go to Addis Ababa. At the time I heard the word Addis Ababa, didn't know where it was; I thought it was not far (and I was also eager) and agreed to her idea. Finally we entered one of the buses without being noticed by the driver assistant. We then managed to arrive at Addis Ababa. I was cheated and betrayed; but it was too late when I understood her intentions. I was only 8 years old and loved to play with kids of my age; that was why I agreed to travel with a girl I never knew before. To return to Dukem, I didn't know how."[5th participant]*

Abuse and ill-treatment at home were precipitating events for homelessness among the participants. Here are what some of them said:

*"While I was herding cattle one day my arm was broken (I was less than 7 years old as I remember). Life was bitter for me then and one day I disappeared from them."[1st participant]*

*"I decided to live on streets than with my mother because my stepfather was abusive to me. Besides, my mother had hypertension and I didn't want her die from the stress."[3rd participant]*

Related to dissatisfaction of life at home, the other related immediate reason was quarrel or conflict with family. For instance, quarrel or conflict of one of the participants with the head of the household resulted in her homelessness. She said:

*". . . . . . . . ... I started living with my sister; I worked during the daytime and went to my sister's home in the evening. However, after a while my sister started throwing intimidating and degrading abusive words at me which I couldn't tolerate. Finally, I left her house to live on the streets."[9th participant]*

For some participants, migration was the immediate cause for homelessness. Some of them said:

*"I came to Addis Ababa with the hope of getting job there; but I had no place to stay and started living on the streets thinking it would be for short time."[8th participant]*

*"One of my friends, who knew Addis Ababa, told me about the work opportunities in the city. Hoping to get better paying job and better life, I decided to go to Addis. I took my youngest half-sister with me, and with the company of my friend, I travelled to Addis. The deal with my friend was that, we were travelling to get better job, but because I didn't know the Amharic language, I had agreed to live on the streets till I was able to speak the language."[10th participant]*

Death of parents was another immediate reason for homelessness among the participants. Death of both parents, especially when the parents are the source of livelihood, resulted in disruption of all family and results in homelessness of children.

*"I ended up on the streets because my parents died. I went into street life one year back when my father died; my mother had died earlier."[10th participant]*

The other immediate cause for homelessness among participants was illness stigma. Some forms of illness were cause for stigmatization in the communities of the participants. Mental and neurological illnesses, developmental retardation as well as epilepsy were some of the sources of stigmatization. One woman reported her experience like this:

*"Two years after I was married I delivered a male child who is now 3 years old. After I was pregnant, my husband's parents forced me to leave for fear that I would die inside their house from my illness; I had epilepsy. According to local myth, epilepsy would be dangerous during pregnancy. They forced me out of their home. Then I decided to go to urban area and get holy water treatment while I raise my baby. Then I went to a town and started living on street for 2 years now."[4th participant]*

One participant was actually born into homelessness; that participant found herself on the streets ever since she remembered. She explained her life history as follows:

*"When I first became aware of my being, I found myself on the street. I don't know my parents; I was raised by street youth who found me abandoned. I am now 19 years and I lived all my life on the streets. It is only recently that my friends (family) of the streets told me they first found me abandoned inside of a garbage container in the City of Adama; they said they went to the garbage container to look for thrown food items. They also told me they decided to raise me and prevented rich people who wanted to take and raise me; they said they told those rich people who requested to take me that I was daughter of their friend's."[6th participant]*

Sexual assault, which includes attempted or actual rape, was another immediate cause for becoming homeless. One participant said:

*". . . . . . . . . . . When I was 6 years old, my aunt took me to Hawasa and raised me. My parents were still alive; the reason I came to my aunt's place is that my aunt had children of my age and I was supposed to play and enjoy with them. . . . . . . . . . . . . . one day my aunt's husband tried to rape me. Following this incident, after living 10 years with my aunt, I took all the money I had and escaped from her to come to Addis Ababa."[15th participant]*

Unwanted forced pregnancy was the other immediate cause. Some participants had lost their opportunity to working as housemaid or a waitress due to forced pregnancy. One of them described it this way:

*"While I was working in the restaurant, I met the barista who is also son of the owners. He was alcoholic and one day he came drunk and raped me after which I became pregnant. Once my pregnancy status was known by the man and his family, they fired me; the man who made me pregnant refused to help me or his offspring. Later, I gave birth to a baby on the street."[7th participant]*

Some participants joined the homeless life because they wanted more freedom. Some of the participants thought that their parents or care givers were too controlling. Those participants assumed that they will achieve the freedom they wanted by living on the street. One participant described:

*"One good man found me and took me to his home in Dukem and raised me like his own daughter. He sent me to school and I finished till grade 3. But his wife . . .; she was very*

*controlling and I didn't like that. I decided to go to Addis Ababa where I didn't know what awaited me."[8th participant]*

Some participants became homeless due to incurable illness which resulted in loss of hope in them. One participant described it like this:

*"My daughter was getting sick very frequently; finally, I was told she had HIV/AIDS. I also got tested and knew I was positive for HIV. Because of this I lost hope and started living on streets."[12th participant]*

Lack of access to affordable housing caused some participants to become homeless. One of them explained her experience as follows:

*"Since the cost of rental house in Addis is too high for me, I even have tried to live in a smaller town near the outskirts of Addis; but I couldn't afford that one either. . . . . . . . . . Finally I lost all my hope in the world; that is when I totally started living on the streets."[12th participant]*

## Way of life for the homeless woman

The way the participants lived while they were homeless differed from one participant to another. The possibilities for how women would live once they become homeless were identified and summarized in this section.

Begging for money, for food, or for other things necessary to life was almost an inevitable experience for them. The participants admitted that the people of Addis Ababa are so generous; and the daily income respondents collected by begging could be to as high as 1000 ETB per day sometimes. That means, if they collect this kind of money regularly, their monthly income could actually be higher than the gross salary of the highest paid public servant in Ethiopia. The following responses were given by participants of the study:

*"On the street I live by begging from people. When it gets hard to get money by begging, I get leftover food ("bulle") from the hotels. I am grateful to the residents of Addis Ababa; they are kind people who give us money and other things generously."[3rd participant]*

*"We started living below a bridge at the center of Addis Ababa. I beg on the streets to get money. We sometimes eat leftover food from the hotels ("bulle"), and other times we buy low cost food by the money we have."[5th participant]*

*"By begging I get as high as 1000 Birr per day. I also have some savings."[14th participant]*

Some participants worked on temporary jobs while they were homeless. When the situation was not as favorable, they complemented their income by begging. The commonest reported job was working as waitress in small bars which sold alcohol, including culturally brewed drinks. Bed-making in hotels and cleaning hotel utensils was another employment opportunity for them. Some of the participants sold small items on the streets; similarly some sold psychoactive substances to other homeless people. Next were statements provided by participants:

*"I occasionally worked in some jobs like rental bedrooms while living on the street. I also was engaged in the trade of 'amag' ('amag' is a code name given by street youth to a commercially available glue chemical used as inhalant substance by street people); I buy the merchandise from the shops and retail it to those who want it on the streets. I get a substantial amount of money from such trade."[6th participant]*

*"Intermittently, we get temporary jobs of washing glasses in hotels; with the money, we buy food items and we cook food. Whenever we couldn't find job and become short of money, we beg from people. I was able to secure a job as housemaid; but I was asked to leave my little sister on the streets since she couldn't be hired together with me. This idea was unacceptable for me and I refused to accept the offer."[10th participant]*

*"I do some paying work at times, and live by begging from people at other times."[12th participant]*

Some participants had no form of shelter throughout their life as homeless. Such participants slept on streets and moved around from one street to another during unfavorable weather or situation such as during the rainy season. Here is what a participant said:

*"We live and sleep on the street and below the bridges of Addis. We don't make any temporary shelters of canvas or plastic because the police would soon damage them."[6th participant]*

Other participants had some form of shelter. The shelters were make-shift small shelters made of canvas and plastic or textile. A participant reported the following:

*"After we arrived at Addis, we started living on the streets by making a canvas shelter."[10th participant]*

Some participants had life which involved getting in and out of the homeless situation. For instance, some participants temporarily lived in rented house with the income they get by working or by begging. Others were temporarily married to men who provided them home as long as they stayed in the marriage. The following responses were given by participants:

*"I met a man while I was working in the bar; he was a regular customer of ours and he was a long truck driver. I started to live with him like married couple for sometime; . . .. I delivered a baby by him."[9th participant]*

*"After I went out to live on the streets, I occasionally was able to rent a house for sometime duration."[12th participant]*

Some participants were involved in commercial sex work during their life as homeless. One of them said:

*"When I was 19 years old I started to do prostitution, despite the fact that prostitution was the last thing I wanted to do."[1st participant]*

Some participants attended school while they were homeless. This was possible for them with the help of other homeless people. One woman said:

*"The street guys caused me to attend evening school; I attended evening school till grade 4."[1st participant]*

Living in groups was a form of living arrangement for some participants. Such living arrangement helped them support each other at times of adversity. One woman reported:

*"We live in groups; each group is like one family. Some of those who live with us are small children. We usually found them wandering around after they were lost from their parents; some*

*of them didn't even start to talk when we first found them and couldn't tell who their parents were and where they lived. We mix them with us and raise them on the streets."[6th participant]*

In some cases participants opted to form partnership with homeless male. This kind of living arrangement helped them to get support from their male partner. One participant said:

*"Five months later, I started a romantic relationship with a man; we still live together. When we were living on the streets we ate 'bulle' whenever we have no money."[15th participant]*

## Positive and negative aspects of life for the homeless woman

Participants were able to identify positive experiences they had in their homeless life. Mutual support and social life within the community of homeless people was described most frequently by them. Homeless people supported each other with supplies, emotionally, and provided protection to one another. The following responses were given by the participants:

*"The positive side of street life is that there is collaboration among street people; we share and eat together. There is love among those who live together on the streets. We care for each other and this gives me happiness. If one of us is attacked by anyone, we come together and defend. I must say, generally, I enjoyed living on the streets: we loved each other, we shared whatever we had among ourselves, and supported each other. For example, if one had no cloth, then he/ she was likely to get from one of the friends in the group. One would not be disgusted by the other; for example, if one bled from injury, another would secure the bleeding with bare hand without worrying about HIV infection."[5th participant]*

*"The positive side of living on the streets was that we loved each other, supported one another, and shared whatever we had. If, for instance, I had 2 Birr and one of my friends had none, I would give 1 birr to him/her."[6th participant]*

Some participants positively reported the presence of freedom in the homeless life. By freedom the participants meant that they don't have to be controlled by their parents or relatives who provide them. Participants felt they could do their will when they are by themselves. Here is what one participant said:

*"The only thing I can mention positively about street life is that there was freedom; it is not like living with parents. I felt I was free and could relax as I wished living on the streets than otherwise; for example, if I lived at home with my parents, I wouldn't be able to use psychoactive substances."[8th participant]*

The participants acknowledged the charitable people of Addis Ababa. The existence of kind people who provided them with useful supplies was their positive experience. One participant said:

*"I can't perceive any positive aspects of street life; what I can witness positively is about the people of Addis Ababa, because even if they knew we were addicted they gave us money."[9th participant]*

The majority of responses, however, were about the negative aspects of the homeless life. Participants described that the negative experiences of homelessness outweighed the positive

ones. Sexual assault was a part of life for them. Many of the participants reported they experienced sexual assault, including rape. Others reported witnessing sexual assault as it occurred to other homeless women. Most of them reported they lived a life always haunted by the possibility of being raped. Participants described that some homeless women collaborated with males to arrange sexual assault on fellow women who are also homeless. They did this in return for incentives such as supply of psychoactive substances. The following responses were given by participants:

*"I was worried about the possibility of being raped; I particularly was worried that my daughter could be raped. Due to my fear of rape of my daughter and myself, I slept during the day and stayed awake during the night to protect my daughter and myself from rapists. . . . . . .When I lived on streets I was chased several times by men who wanted to rape me; in those incidents, I saved myself from being raped by shouting and calling for help. One day, however, I was overwhelmed by sleep and my daughter was raped; they also raped me after they covered my eye and packed my mouth. My daughter was only 5 years old."[7th participant]*

*"I was usually unable to sleep because men tried to rape me; I had to stay all night awake to protect myself against rapists"[13th participant]*

Abusive police officers were a source of negative experience to the participants. They chased them from the places where they slept, insulted them, beaten them severely, destroyed their shelter, and falsely accused them of wrongdoing. Participants explained as follows:

*"The other challenge is that the police sometimes came and wake us from where we slept. The police at times accused us for theft even if we didn't steal anything; I remember of an incident when we were beaten for wrongdoing which we actually didn't commit."[6th participant]*

*"The police come to where we sleep and hit us even without asking about our problems and the reasons why we were there. They once hit me together with my small baby daughter; they also insulted us in the most offensive derogatory words."[9th participant]*

The participants had other grievances and worries which affected them negatively. These included grief from memory and concern about their parents, children, and worries about their children being possibly stolen on the streets. Here are some of their responses:

*"The memories of my mother and my son haunted me frequently, and made me suffer. Especially, the main worry I had was about my mother. I feared she could die from her hypertension. She got angry easily and her blood pressure increased."[3rd participant]*

*"Another thing I was worried about was that my daughter could be stolen; I had heard children were stolen for their organs."[7th participant]*

Feeling of shame from the act of begging was another negative experience for the participants. They said:

*". . . . . .. It is very shameful to stand before people asking for money."[3rd participant]*

*"People despised me and sometimes spat on me which was intimidating and degrading."[8th participant]*

The participants suffered from harsh weather conditions while living on the streets. They described the experience as follows:

*"The challenging or the difficult part was that there was rain and floods during the rainy season; this became a big challenge to us. During one of those rainy days, we could be forced to spend the night wandering in the city, and not be able to sleep."[6th participant]*

*"I once was sick and unable to walk from the cold weather's effect on my body. The street life had become horrible to me."[13th participant]*

Hunger and food shortage was a frequent experience for the participants. They described this negative experience as follows:

*"Sometimes I would be forced to spend all day without eating for lack of food"[5th participant]*

*"There were days when I had nothing to eat and went to bed empty stomach"[13th participant]*

Their belongings were stolen from them and this was painful for them. Participants said:

*"Thieves also would come during the night while we were asleep"[8th participant]*

*"My daughter's clothes were stolen"[12th participant]*

Drunken people disturbed while participants slept on the streets at night. They described it as follows:

*"There was also possibility that people who were drunk would come and disturb us. It was boring to live on the streets."[8th participant]*

*"The hard and challenging aspect of street life outweighs. . ... The following events occurred almost on daily basis: A drunken person would come and disturb, or insult me."[11th participant]*

The personal feeling of being stigmatized due to becoming homeless was also a negative aspect of the homeless life. Participants mentioned the following:

*"People saw us like crazy people and due to that they were afraid of us. Because of this I felt that I was different and felt isolated and lonely."[9th participant]*

*"The challenges I had to bear during my life on the streets were: The people who had known me before I was on the streets despised me and intimidated me whenever they saw me on streets."[12th participant]*

Participants were economically exploited, especially by male partners. One of them described her negative experience as follows:

*"Whatever I earned from commercial sex work, I gave it to my new boyfriend; he was so convincing when he spoke, I always trusted him. One day I told him that I hated working as prostitute and that I wanted to get out of it; he disagreed with this idea and even told me to stay away from him thereafter. Then it became clear to me that he was after the money that I earned from commercial sex work and didn't love me."[1st participant]*

Sexual exploitation was also reported by participants. One of them said:

*"I met someone on the streets who promised me we would live together and I agreed with his idea. Then we started living and having sex together on the streets; I became pregnant in the meantime. However, he abandoned me after he knew I was pregnant."*[13th participant]

Lack of supplies was a frequent impediment for hygiene among the homeless women and participants felt bad about it. One of them explained:

*"We hardly could afford to buy even hygiene soap in most days."*[8th participant]

Some people from among the homeless people sometimes forced the participants to use psychoactive substances and this was usually painful to them. One participant gave the following narrative:

*"Some people urged us to be substance users."*[12th participant]

## Ways of coping with stress and adjusting to life as a homeless woman

The participants described their ways of coping and/or adjusting to homeless life. They used psychoactive substances to cope with their daily stressful experiences.

*"I used psychoactive substances as pastime, and to cope with my feeling of loneliness."*[1st participant]

Some participants formed a sexual relationship with male partner in order to feel safe and protected. They explained:

*"I also was able to have a boyfriend from among the street boys which was important for me to be protected from sexual violence on the streets from the street boys. . . . . . . . . .Because I had a boyfriend from the street boys who is one of them, the other boys wouldn't think of me sexually. Otherwise, if I were alone I would have the risk of being gang-raped by 4 or 5 of them at a time; this happened to other women on the street."*[1st participant]

*"One time I decided to 'get married' on the streets and started a relationship with one of the street boys. The main reason for me to do that was to get protection from sexual assaults. I also intended to give birth as soon as possible; I made this decision because a woman on the streets, if she was in a relationship and/or had a baby, the men were unlikely to assault her and even tended to protect her. I got pregnant with such an intention now."*[6th participant]

Some participants tried to make their homeless life easier by forming good rapport and using good social skills with others.

*"I was able to cope with street life because I was able to create and maintain rapport with anyone, particularly those who lived on the streets with me. I handled challenges politely; if I got irritable, those guys were likely to be more violent and abusive. If I became polite and gentle with them I knew I could live with them peacefully. I handled menacing tendencies and approaches of some of the street boys carefully, wisely avoiding them and without becoming confrontational."*[1st participant]

Participants tried to conform to the street culture so that they could be accepted by the others.

*"To cope with the challenges from men, I tried to conform to the street culture and tried to be like them; I did what they did."[3rd participant]*

Some participants dressed and behaved like male in order to escape sexual assault.

*"I was challenged by males who wanted to take advantage of me even after I changed living area. I then took a wise decision to cut my hair short and dress like a male, so that they would think I was male. This strategy worked well for me."[5th participant]*

Some participants lived close to churches to get protection, as well as to benefit from donations of money and items from people who come to worship.

*"The reason why I was not raped was the fact that I spent day and night around the church."[11th participant]*

## Reasons why homeless women stay homeless

Some factors kept the women homeless by making exit to secure housing difficult or impossible. Some of the factors are listed below.

Participants described that they were unable to find affordable house; because of this reason they were not able to exit homeless life. They explained:

*"If I got the chance, I would like to get myself out of the streets. The main impediment for me not to get out of street life was that the cost of rental house was too high; otherwise I was confident that I could work and earn my living. The one thing I needed support for was getting affordable house. . . . . . .."[11th participant]*

*"If I got the opportunity, I wanted to get out of street life. The main obstacle I had was the cost of rental house. If someone supported me to get a shelter, I would see no problem in getting into some kind of work and support myself and my children."[12th participant]*

Some participants had the feeling of shame about going back to their families. They felt it would be an embarrassment to their family, as well as to themselves, to go back home after having had homeless life.

*"The main reason why, at that point, I didn't want to go to my parents was that I didn't want them to see me in that condition carrying a child whose father was unknown, and addicted to substances."[8th participant]*

*"What held me back from going to my place of origin where I had relatives was, in my community, it would be an embarrassment for myself and my family to have been an unmarried woman with children. People wouldn't give me the peace of mind and I would suffer from rejection. My most disturbing worry was to not lose my children."[12th participant]*

For some participants there were unfavorable situations back home preventing their return. The very reasons which caused them to leave home to become homeless still existed.

*"If I got the chance, I would have wanted to get out of street life. I would want to start working and support my mother and send my son to school. My mother would get old and weak from*

*time to time; she had a coffee growing farm around Harar. However, I didn't want to go to her because there was a stepfather; my father had been dead. Because of the presence of a stepfather at home, I didn't want my mother to suffer by my presence in the house; my stepfather never had had good attitude towards me."[3rd participant]*

*"I could not go to my place of origin because there was no one I could rely on. My father lived with support from charitable folks; there was no one who could support me."[11th participant]*

Some participants had no place to call home and they felt there was no place for them to go.

*"One day, I was so bored by and hated street life I decided to be involved in crime so that they could take me to jail. I considered jail as an asylum from street life. Then I confessed for a theft which I didn't commit; for that I was sent to prison for 3 months. I would have wanted to get out of street life if I got the chance; but there was no place I could go because I had no home."[6th participant]*

Some participants could not exit their homeless life because of addiction on psychoactive substances.

*"Once I became a street girl, I couldn't, and didn't want to go back home because of my addiction. I felt it was hard for me to abandon the substances."[9th participant]*

Some participants reported that they simply had no transport money to pay for journey back to their home or native areas.

*"If I get the chance, I would want to get out of street life because: I felt street life was boring; I wanted to work, and be self-sufficient. However, to go back to home at Wolaita, I didn't have enough money for transportation."[10th participant]*

## Discussion

This is a study conducted in a low-income country, where there are no regular shelter facilities or services meant for the homeless. The study was conducted in a country which had cultural norms which gave women a lower and submissive status and where child marriage was widely practiced. Our study differs from previous similar studies conducted in Ethiopia because it is more comprehensive. This study has tried to explore with much detail the life history of homeless women starting from early childhood, through events leading to homelessness, to the possibilities and the means of survival of the women, to the dynamics of positive and negative experiences they had, as well as their ways of coping with challenges.

The study identified that most of the causative factors for women's homelessness among the participants were similar to findings of previous studies. In this regard, domestic violence, poverty, and lack of access to affordable housing which previous studies had found out were also identified in our study [8,18,23,25,27]. However, our study differs from previous studies in that it defined homelessness to include only those sleeping in designated public spaces or shelters, or those who were 'roofless'; as such, the study included women under the extreme form of homelessness. This could mean that the factors causing homelessness in our participants were likely to be in their severest forms. In that case, the threshold of tolerance of our participants to socio-economic and psychological stress seems to be high. Findings of our study conform to those of studies conducted in big cities in Ethiopia. The factors associated

with causation of homelessness among women in previous studies in Ethiopia were also identified in our study; those factors were migration, urbanization, poverty, abuse, escape from child marriage, unemployment, death of one or both parents, divorce of parents, pregnancy out of wedlock, and abandonment by spouse [14,15,29,30]. This shows the existence of factors which are similar throughout Ethiopian cities. We, however, identified more factors in the current study such as deception, family quarrels, search for more freedom and stigma from illness.

As can be seen from the analysis results, the early lives of the study participants were not satisfactory to them; for some participants, their early life experiences were even harsh and unacceptable. Therefore, it can be noticed that homelessness among the participant women was not determined only by the occurrence of triggering events which precipitated homelessness. Instead, there were situations in their lives which created fertile grounds for homelessness. Those fertile grounds indicated the predisposing factors. The cultural, social and legal disadvantages of women which existed in the cultures of the participants were identified as predisposing factors in our study. For instance, there was a cultural occurrence of child marriage in rural parts of Ethiopia. Some of our participants were given to such marital commitment as early as the age of 5 years. While such marital commitment was made between the parents on both sides of the married, due to issues related to dowry and other disagreements between the parents the life of the small girls became too difficult to bear leading to homelessness. The other culturally-sanctioned predisposing factor for homelessness in this study was history of commercial sex work. Since prostitution was extremely stigmatized in the culture of the society of the study participants, those who had the exposure were ashamed to stay with parents/families or their communities for fear that they or their families would be isolated and humiliated. Finally, they were forced into the homeless life. Difficult marriage was another important issue which we find worth discussing. In the culture of the participants, a woman in a difficult marriage was expected to tolerate hardship and maintain her marriage. For a woman in that kind of situation, especially if she was economically dependent, homelessness was likely to occur at some point; this is because going back to live with her family after separation with spouse was likely to result in humiliation to her and her family. Therefore, intervention measures that need to be taken at the societal level should address the above mentioned issues.

The life challenges which our study participants experienced were similar to the findings of previous studies. In our study homeless women experienced sexual assault, substance abuse, psychological distress, as well as labels and stigmatization; such experiences were also identified in previous studies [18,19,23,28,31,32]. Our findings also conform to similar studies conducted in Ethiopian cities. In previous studies homeless women were found to be vulnerable to rape, physical abuse, emotional abuse, extreme poverty, hunger, physical safety concerns, lack of safe drinking water, poor sanitation and prevailing diseases; our findings were in agreement with those findings from Ethiopian studies [14,15,29,30].

As a means of mitigating abuse and coping with street life, our respondents used various strategies. Using marriage as protection against sexual violence was identified in our study, a finding similar to a previous study conducted in Addis Ababa [14]. However, despite the use of forming partnerships with men on the streets, our study showed homeless women were economically and sexually exploited by their male partners resulting in psychological trauma to them. Using marriage for protection also resulted in the possibility of exposure to unwanted pregnancy. The likelihood of getting an unwanted pregnancy, either from rape or from marriage was considered high. A previous Ethiopian study showed that one-fourth of street children were born to street women, making the second generation on the streets [15]. This shows how women homelessness sustains and complicates the problem of homelessness as a whole

by making homelessness intergenerational [20]. Much of the income of the homeless population in Addis Ababa came from begging on the streets, with casual work making another source of income [39], a finding similar to our study.

One prominent finding of our study showed that respondents benefited from charitable provision of money by the residents of Addis Ababa. However, this behavior of the residents of Addis Ababa could have been aggravating the situation by encouraging more women to become homeless. The painful truth identified from our study was that some of our participants earned income so remarkable, even the most hard-working professional people wouldn't be able to match their daily income. This could be a factor perpetuating homelessness, making exit from the situation less likely. The other perpetuating factor identified in our study was the lack of organized and sustained support for homeless women to enable them to move to secure housing. In fact, recommendations which were based on the existing literature indicated the need to provide homeless women and girls with assistance in obtaining housing, feeling a sense of community, flexibility in housing programs, and having options and choice in housing selection [21].The lack of such assistance to the homeless should be taken seriously; in fact, institutions could help to make better use of the donations from the people who are willing to support the disadvantaged.

Several of the participants indicated that they could work and support themselves except for the fact that it was difficult for them to afford for a housing in the city. In fact, finding affordable housing had become a daunting task in many of the cities in Ethiopia. The current study has limitations within which the conclusions should be interpreted. One limitation the study is that the researchers' previous understanding about the topic could have influenced the process of the research and the conclusions to some extent. However, as much participant narrations as possible were provided to give the reader better understanding. The sample of 14 was considered low and also not representative of all homeless women in Addis Ababa, but the aim of the study was in-depth understanding of the situation and generating ideas for further evaluation rather than representativeness. Therefore, the conclusions need to be understood in the context of these limitations.

## Conclusion

In conclusion, the pathways through homelessness in the study participants could be formulated into an explanatory model. On the background, difficult or unsatisfactory living conditions of women in the community produced fertile ground for the occurrence of homelessness. On top of those predisposing factors, occurrence of additional triggering life events served as immediate cause of homelessness; such precipitating life events made the misery of the women surpass the threshold of tolerance, and made homelessness inevitable. Once homeless, there were positive and negative experiences and perceptions of the women, as well as the means of coping with the stress of the homeless life. The net effect of the interplay between the positive and the negative, as well as the effectiveness of efforts to cope with stress decided whether the woman would stay homeless or exit out of it. The existence of perpetuating factors finally affected the decision to stay homeless or move to secure housing. We recommend representative quantitative studies to be conducted.

## Acknowledgments

The authors are grateful to the study participants for their willingness and their eagerness to provide valuable and genuine information. The authors acknowledge Mr Fasil Girma, and his staff, for the support provided during the process of data collection.

## Author Contributions

**Conceptualization:** Kibrom Haile, Halima Umer.

**Data curation:** Kibrom Haile.

**Formal analysis:** Kibrom Haile, Halima Umer.

**Investigation:** Kibrom Haile, Halima Umer.

**Methodology:** Kibrom Haile, Tolesa Fanta, Addis Birhanu.

**Project administration:** Kibrom Haile.

**Resources:** Kibrom Haile.

**Software:** Kibrom Haile.

**Supervision:** Kibrom Haile, Halima Umer, Tolesa Fanta, Addis Birhanu, Edao Fejo, Yirga Tilahun, Habtamu Derajew, Agitu Tadesse, Gebreselassie Zienawi, Asrat Chaka, Woyna-beba Damene.

**Validation:** Kibrom Haile, Tolesa Fanta.

**Visualization:** Kibrom Haile.

**Writing – original draft:** Kibrom Haile, Halima Umer, Tolesa Fanta, Addis Birhanu, Edao Fejo, Yirga Tilahun, Habtamu Derajew, Agitu Tadesse, Gebreselassie Zienawi, Asrat Chaka.

**Writing – review & editing:** Kibrom Haile.

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
