## [Decision Letter · Decision Letter 0]

17 Jun 2020

PONE-D-20-15059

Pathways through homelessness among women in Addis Ababa, Ethiopia: A qualitative study

PLOS ONE

Dear Dr. Haile,

Thank you for submitting your manuscript to PLOS ONE. After careful consideration, we feel that it has merit but does not fully meet PLOS ONE’s publication criteria as it currently stands. Therefore, we invite you to submit a revised version of the manuscript that addresses the points raised during the review process.

We look forward to receiving your revised manuscript.

Kind regards,

Mellissa H Withers, PhD, MHS

Academic Editor

PLOS ONE

Journal Requirements:

Reviewers' comments:

Reviewer's Responses to Questions

**Comments to the Author**

1. Is the manuscript technically sound, and do the data support the conclusions?

Reviewer #1: Yes

Reviewer #2: Yes

2. Has the statistical analysis been performed appropriately and rigorously? 

Reviewer #1: N/A

Reviewer #2: N/A

3. Have the authors made all data underlying the findings in their manuscript fully available?

Reviewer #1: No

Reviewer #2: Yes

4. Is the manuscript presented in an intelligible fashion and written in standard English?

Reviewer #1: No

Reviewer #2: Yes

5. Review Comments to the Author

Reviewer #1: Overall, this very interesting and important paper would benefit from an additional revision to streamline and correct errors in the writing, with attention paid to use of words and grammar. An example is a sentence in the Results section of the abstract: “Finally, the presence of perpetuating factors such as lack of affordable house, feeling of shame to go back home, unfavorable situation at home, etc discouraged participants from exiting the homeless situation.” There are several areas of this sentence that read awkwardly. This is the case with sentences throughout the manuscript, including missing words (such as ‘the’ and ‘a’) and extra words, and missing but needed pluralization. Often the word ‘house’ is used when ‘housing’ is the correct form. Consider tense – the background uses both the present and the past, sometimes in the same sentence. The authors’ presentation of their strong argument would be greatly improved with editing.

ABSTRACT

Methods should mention how analysis was conducted. Interviews were conducted ‘with’ 14 women, not ‘on’ 14 women.

Results: Use of etc. in the sentences is not necessary because it is not informative. Everything mentioned about homeless life is negative, however there is a sentence referring to the net effect of negative and positive experiences. It is intriguing because the next sentence refers to whether homelessness is tolerable, so the reader is left wondering about any of the positive experiences in the data. I recognize that this is the abstract and brevity is essential, however a reference to positive without examples such as is given for negative, is a problem.

Conclusion: There is always a danger of overstating the findings in a qualitative study. In this case, use of the word ‘determined’ in the first sentence does indicate overstatement. In the second sentence, however, the participants are referenced so they authors are not generalizing beyond this small study, which is appropriate.

MAIN TEXT

BACKGROUND:

This section is quite lengthy. It includes an excellent review of literature on the various relevant topics, however some of the studies overlap in their findings, and the points are therefore repeated. Also, there are so many topics covered, even within a single paragraph, with each sentence starting a new topic. Due to this, the section reads more like a page of disparate notes than the presentation of a seamless narrative about this important problem.

METHODS:

-Since the participants know some of the authors as volunteers, what was the procedure for ensuring they understood that in this case the authors were acting as researchers, not volunteers?

-Great to see that purposive sampling was used.

-For the sentence that includes “adult women respondents who had rich experience of the homeless life” it would be better to say they could provide rich data about their homeless life experience.

-Rather than respondent, as is used in a survey study, participant is more appropriate for qualitative research.

-The use of grounded theory and how phenomenology guided the study needs to be explained.

-The first paragraph actually includes information that is or should be in the subsequent subsections. For example, sampling has its own section below.

Setting and sampling strategy:

-It is unclear what ‘own-account’ workers are. Does this listing account for informal sector workers?

-Better to have setting as one subsection and sampling as another as they don’t really go together.

-Since the entire sample came from the one shelter, I’m curious to know more about the shelter in terms of what it provides to the women, if their stay there is time-limited, and what the accommodations and services are like.

Data collection:

-Rather than ‘data collector’, interviewer seems more fitting.

-Development of the interview question guide is missing. This is particularly important to discuss since grounded theory method starts from the very beginning, and iterative modification of the interview guide is an integral process. The one sentence saying that it was an open ended interview guide does not provide enough information. What were the topics? How were they decided upon? How did grounded theory and phenomenology inform development of the questions?

Data analysis:

-The paper needs an entirely different data analysis section. Grounded theory is a very specific type of qualitative research approach, and it’s clear that the authors did not use grounded theory. That’s fine – grounded theory is only one of many approaches. But they should not state that they’re using it if they are not. I suspect that the reason this term was chosen is that the authors did not have a stated theory underlining the study, and they used inductive coding without an a priori code book developed for the coding process. However, the authors should just describe exactly HOW they arrived at their analyses, themes and interpretation – and not be concerned with a label such as grounded theory, which just doesn’t fit their process.

-A theme cannot be identified in advance; themes emerge from analysis of the data. Topics are identified in advance and incorporated into the interview guide, which is what I assume the authors did. That themes were entered into the variables section of the software doesn’t tell the reader anything meaningful – what exactly does this mean for the analysis process? Did the themes emerge by carefully reading all the transcripts before coding? That is the only way that subsequent coding for themes could happen, but the authors do not describe this as their process. It could be that the terms themes and codes are being used erroneously or in a confusing way in this section, leading to the lack of clarity.

RESULTS

-What does it mean that during analysis major segments were identified?

-Themes are now called categories in this section – It seems that the authors are misunderstanding what themes are in qualitative analysis. Often there are topical categories that don’t quite make it to a theme designation, and that’s fine – just not alright to mix it all up. -The entire first paragraph of the results section is confusing; but in any case it is about analysis and not results so it can be deleted.

-Usually results sections begin with a description of the participants – this was in part included previously in the paper, and could be moved here.

Predisposing factors:

-Again, analysis process is included here though it belongs in the analysis section. Though the sentence is unclear in its meaning: “Under this category, the thematic areas were identified and grouped into codes to give the existence of the following factors.”

Positive and negative experiences:

-After briefly describing the positive experiences, the authors go back into the negative experiences, some of which were already described in previous sections. I realize the subsection is called positive and negative experiences – but these negative experiences should instead be woven into previous sections, or negative experiences should have its own subsection. I would say the issues regarding sexual assault should have their own subsection among the negative subsections. Also, given the situation with women having to stay awake at night to avoid being raped, it would be helpful to know if the women who described having a supportive street family are not the women who were worried about being raped, and if so the importance of the street family can be emphasized as being protective, besides being enjoyable.

General issues in the results sections:

-The quotes strongly demonstrate that the interviews generated rich data about these heartbreaking issues. The introduction to each section of quotes is important for setting up the reader’s awareness of what the quotes are meant to represent. The authors should review each of these paragraphs to ensure that enough information is provided, or that the sentences each provide additional information. For example, the following three sentences really all say the same thing so are not all necessary: ‘The other immediate cause for homelessness among participants was illness stigma. Some forms of illness were cause for stigmatization. Illness stigmatization may come from some types of illnesses.” Instead of repeating the same information, it would be helpful to know additional information such as what kinds of illnesses cause stigma. The quote is about epilepsy but the authors imply that there were a number of illnesses discussed in interviews that cause stigma. Another example where the two introducing sentences say the same thing is: “Some respondents joined the homeless life because they wanted more freedom. The participants who made such decision assumed that they will achieve the freedom they wanted by living on the street.” More information about the nature of the freedom they were looking for would be helpful. The quote can only describe one instance of this but the authors say “some respondents” so information about all the types of situations would better explain this theme. A necessary use of the intro paragraphs for each set of quotes is to summarize what is in the data on the topic, and then illustrate one or two significant elements of that.

-Some of the intro paragraphs give a good amount of information, such as the one on page 15 about working temporary jobs. This could be a model for the rest of the topic paragraphs.

-The following kind of sentence, which ends the paragraph I’m referring to, is not necessary: “Next are statements provided by participants:”

-Be certain that what the intro paragraph says isn’t just exactly repeated in the quote. Either put more into the intro paragraph, or simply don’t use the quote.

-An example is: “Other respondents had some form of shelter. The shelters were make-shift small shelters made of canvas and plastic or textile. A respondent reported the following:

“After we arrived at Addis, we started living on the streets by making a canvas shelter.”[10th participant]”

-Review the entire results section to ensure that the same topic does not appear more than once, even if it is describing more than one theme. Better to find a way to combine. An example is forming sexual relationships that appears several times under different subsections.

-Another theme that appears could be expanded is their initial trust in older sisters and subsequent negative experiences living with them), and their responsibility for younger sisters.

DISCUSSION

The authors include excellent points in the discussion. It is difficult, however, to follow the arguments due to the style of writing. This discussion is very long, and it would improve the paper to consider streamlining the recap of the findings so that not so many details are included, but the main points and comparison to the literature are highlighted. This section also requires editing for phrasing and grammar.

Reviewer #2: This paper gives a glimpse into a major social challenge common to many big cities. Particularly the result section is engaging like a good novel. But it is about hard realities!

The authors have used ground theory, not indicating any theoretical framework. They also refrain from defining “homelessness” and start off with a small selection of women sleeping in a “design public space”. To see what this implies of bias or strength I would appreciate some elaboration of what this shelter is, how long the women had been living there and how they were recruited.

There are some other points that I could like to see elaborated: Successful begging might have a specific cultural background? There are “peculiarities regarding gender role of women”, which? Much is said about networks and support among the homeless, but (at least some decades ago) there was also a unique network of elderly women who took care of young girls who came to Addis.

I liked the authors’ search for a “tipping point” that got the women to take to the streets. This is an approach that has proven fruitful in studies of health seeking behavior and defaulting from recommended treatment. Would it be possible to see links between specific risk factors and types of tipping point events?

When the authors end their conclusion by saying: “We recommend representative quantitative studies to be conducted.” I could not agree more. The present study gives very valuable understanding of the situation for the homeless women, but only quantitative data can move decision makers to take action. And I think the qualitative data here gives good basis for calling for goal directed quantitative studies.

6. PLOS authors have the option to publish the peer review history of their article (what does this mean?). If published, this will include your full peer review and any attached files.

Reviewer #1: No

Reviewer #2: Yes: Gunnar Bjune

---

## [Author Response · Author response to Decision Letter 0]

23 Jul 2020

To editor:

Dear editor, thank you for the interest you showed in our work, and for the strong consideration. We, therefore, have relied as follows the issues you requested us to address. We also have given point-by-point response to reviewers’ comments in a separate file attached as ‘Response to Reviewers’. Besides, we have attached two other files with amendments to the manuscript after receiving the comments of reviewers and made corrections accordingly. Regards

a) About not being able to share data

The reason why we are not able to share data publicly is because doing so will compromise participant privacy, and violates the terms of consent of the participants made before the interviews. 

We have also reviewed the PLOS guidelines about issues of data availability and unacceptable data access restrictions section and we assure you that our decision is in alignment with the criteria included in the Journal’s criteria. The specific issues of privacy concern are as follows:

• The data include life histories of the participants with repetitive mentions of physical address and places they have been at

• The data include information that, in combination, makes identification of the participants possible

• Data is collected from a small group of vulnerable population group

• Indirect identifiers such as sex, ethnicity, places or origin and previous places where they lived that may risk identification

• Most of the data includes sensitive information of this vulnerable population group.

 We assure you that the reason we are not able to share data publicly does not qualify for any of the unacceptable data access restrictions listed in the Journal’s guideline. 

Thank you again

To reviewers:

First of all, we the authors would like to appreciate the interest you have in the topic. Thank you also for the value you saw in this work and the hard work invested on it. We want you to really understand that we like the way you gave your comments; very specific and critical. Therefore, we provide our point-by-point responses to the comments you gave and the corrections we incorporated to the manuscript.

1. Abstract 

• In the ‘methods’ section of the abstract ‘with’ was written instead of ‘on’ as you suggested. Shown also by track change.

• How data analysis was conducted has been included in the abstract section. This is shown by track changes in the revised manuscript.

• In the ‘results’ section the word ‘etc’ has been deleted as you suggested.

• Again in the ‘results’ section, a sentence stating positive experiences has been included to make better meaning. Finally it reads: “Despite mentions of positive experiences of homelessness such as mutual support and good social life within network of homeless people, the net effect of the interaction between negative and positive experiences of the homeless life----“ The amendments are also shown by track changes on the file named ‘manuscript with track changes’.

• In the ‘conclusion’ section the word ‘determined’ has been changed by ‘resulted in –the occurrence of homelessness—among the participants as shown by track changes.

2. Main text

• About the ‘background’ section, to be honest, we believe it has good flow and single concepts are discussed in each paragraph. Even if it looks lengthy, we believe it is not boring and provides a summary of the literature in one section. We prefer to keep this section intact.

• In the ‘methods’ section, the following corrections have been made:

About the sentence stating some of the researchers were also volunteer workers in the shelter, the statement was qualified and re-written as follows: “The second and sixth authors were volunteer workers of the shelter and they had established prior relationship with the participants; however, whenever those volunteer workers were involved in the research activities they said so to the participants and assured them that their participation in the research activities was fully voluntary, and without any negative or positive consequences upon the services they received in the shelter facility.” Also indicated by track changes on the revised manuscript.

The statement written as “adult women respondents who had rich experience-----“ was corrected and re-written as “ adult women participants who could provide rich data about-----“ as you suggested.

About the use of ‘grounded theory’, this statement was included in the ‘method’ section to indicate that the analysis used techniques from those theories in that inductive method was used to suggest theoretical framework for the pathways to homelessness. This doesn’t mean that our techniques were fully loyal to this kind of approach, nor that we used the full approach of grounded theory in our study. Likewise, ‘phenomenology’ was mentioned to indicate that our study analysis included aspects of homelessness as experienced by the participants and themes emerged from those experiences. However, as you suggested, mentioning ‘grounded theory’ and ‘phenomenology’ may not be appropriate or it could actually be confusing because those specific approaches were not used in their full and real sense. Therefore, we have decided to delete those phrases from the manuscript and focus fully on what actually was done during data analysis in our study. 

‘Setting’ and ‘sampling strategy’ have been separated and re-organized as you suggested. This is shown on the file ‘Revised manuscript with track changes’.

The ‘setting’ section has been divided into two paragraphs and the following statements have been added at the end of the second paragraph: “Recruitment of the women to the shelter was from all sub-cities of Addis Ababa and on voluntary basis. The shelter was established and funded by the government and was run in collaboration with volunteer workers. Food, clothes and shelter was provided by the facility, as well as medical, mental and psychosocial services. The beneficiaries stayed at the shelter for maximum of 5 months including the 3 months allocated for vocational training which included hair-dressing, cleaning and tailoring.” This is shown by track changes in the revised manuscript

In the ‘data collection’ section, the term ‘interviewer’ was used instead of the former word ‘data collector’ as you suggested and this amendment has been made to the manuscript.

About development of the interview guide questions, we must indicate this is an explorative study and the questions prepared were general open-ended questions which would help guide the interview, and without limiting the ideas that are to be raised during the interview. We didn’t use the full approach of the ‘grounded theory’ as we replied to earlier comments. The interview guide questions were prepared considering the inclusion of the broad areas of the ‘homelessness pathways’ in the interview. The broad areas and possible ‘future themes’ were anticipated from our previous knowledge and literature review. The interview guide questions are listed below:

1. Please tell me about yourself (you may not mention your name) and your life before you came to the homelessness situation.

2. How did you end up becoming homeless?

3. How do you describe the life of a homeless person?

4. Can you tell me the challenges you face in your life as a homeless person?

5. How do you try to cope up with the daily challenges of living on the streets?

6. If you get the chance, do you want to get out of the homelessness situation?

7. Anything you want to add, you are welcome.

About the generation of themes before data collection, some literature indicate that it is possible, and not an uncommon practice to identify possible themes in advance based one’s prior theoretical understanding, experience or literature review of whatever phenomenon one is studying. In fact, we included specific guiding questions in the interview guide questionnaire with those ‘themes’ in our mind. This doesn’t mean we had the themes before the interviews, but it means we had those possible ‘themes’ under consideration before the interviews based on our previous knowledge and used interview guide questions to make sure those important considerations are not missed. However, the themes and subthemes were identified after the interview. However, we also believe that the critic of the reviewers is legitimate; therefore, we have modified the statement by replacing ‘topics’ for ‘themes’ in that particular context. The data analysis section was re-written to address your concerns; please refer to the file ‘revised manuscript with track changes’.

In the data analysis section, the word ‘theme’ was misused in some statements as you commented, especially the ‘major themes’ or ‘segments’ were inappropriately used. We have made corrections to it by using the word ‘topic or topics’ instead. 

The data analysis we used fits to thematic analysis and we have included a sentence to the ‘data analysis’ section to indicate this if you don’t mind.

• In the ‘results’ section

The first paragraph has been re-written and moved to the ‘data analysis’ section as you suggested. This is shown by track changes in the revised manuscript.

The beginning of the ‘results’ section the characteristics of the participants have been described as you suggested. 

In the ‘predisposing factors’ the sentence about categories and themes was removed.

In the ‘results’ section the word ‘respondent’ has been replaced by ‘participant,’ and the intro statements have been elaborated a little more as shown by track changes in the revised manuscript.

‘Negative and positive experiences’ sub-section shows participants’ attributions of events and their perception of them as either ‘positive’ or ‘negative’. This sub-section shows aspects of positive and negative events and provides opportunity for comparison; as such we think there is no problem with leaving this sub-section as it is. Besides, the topic was identified and categorized during data analysis and it plays a particular role in explaining pathways through homelessness. Therefore, we have decided to keep it as it is.

Sexual assault was not given its own sub-section because it is a negative experience and it doesn’t make sense to give it its own separate sub-section. 

The positive and negative experiences are opposing aspects of the participants’ experiences of homeless life which could happen to any of the participants at different times. The participants who had positive experiences do not see the homeless life as ‘all good’. In fact some of the participants described some experiences as ‘positive’ and other experiences as ‘negative’. Those who had supportive street family also had described ‘negative’ experiences due to other aspects of homeless life; they also worried about the possibility of sexual assault which could come from strangers due to their situation of homelessness. In fact, as was described in the manuscript, some of the participants acknowledged the protective aspects of street family (especially having a ‘husband’), but it doesn’t imply that homeless life is ‘enjoyable’ at all; participants were trying to see some ‘positive’ aspects of the life full of plight.

• The ‘discussion’ section has been edited for grammar and tense. Please check the file ‘Revised manuscript with track changes’ for all the editions. Thank you very much.

---

## [Decision Letter · Decision Letter 1]

10 Aug 2020

PONE-D-20-15059R1

Pathways through homelessness among women in Addis Ababa, Ethiopia: A qualitative study

PLOS ONE

Dear Dr. Haile,

Thank you for submitting your manuscript to PLOS ONE. After careful consideration, we feel that it has merit but does not fully meet PLOS ONE’s publication criteria as it currently stands. Therefore, we invite you to submit a revised version of the manuscript that addresses the points raised during the review process.

Please address the comments of the first reviewer. While the paper is much improved, there are still several points that need further clarification or edits.

We look forward to receiving your revised manuscript.

Kind regards,

Mellissa H Withers, PhD, MHS

Academic Editor

PLOS ONE

Reviewers' comments:

Reviewer's Responses to Questions

**Comments to the Author**

1. If the authors have adequately addressed your comments raised in a previous round of review and you feel that this manuscript is now acceptable for publication, you may indicate that here to bypass the “Comments to the Author” section, enter your conflict of interest statement in the “Confidential to Editor” section, and submit your "Accept" recommendation.

Reviewer #1: (No Response)

Reviewer #2: All comments have been addressed

2. Is the manuscript technically sound, and do the data support the conclusions?

Reviewer #1: Yes

Reviewer #2: Yes

3. Has the statistical analysis been performed appropriately and rigorously? 

Reviewer #1: N/A

Reviewer #2: N/A

4. Have the authors made all data underlying the findings in their manuscript fully available?

Reviewer #1: No

Reviewer #2: No

5. Is the manuscript presented in an intelligible fashion and written in standard English?

Reviewer #1: No

Reviewer #2: Yes

6. Review Comments to the Author

Reviewer #1: The paper flow is much improved and clearer with the changes made by the authors. Following are minor points that should be addressed:

Sampling; insert the word ‘case’ after typical; include a reference for readers who are unfamiliar with qualitative sampling.

Ethics approval: Much of the contents of the consent information sheet is typical of consent forms and so not necessary to include in this paragraph.

Results: in the sentence starting: “On of the participants” the first word should be One. In the sentence “Among those who had at least one children,” it should be child.

Step-parent or step parent? Need consistency.

Since the shelter is described as being for adult women and elderly men – how is it that some of the women had children? Were the children living with them at this shelter? Is it therefore a shelter for adult women and their children as well as elderly men?

Revise the sentence “In some cases participants were opted to form partnership with homeless male.” To: In some cases, participants opted to form a partnership with a homeless male.

In the following sentence, I’m feeling like ‘intimidating’ is not exactly the word the authors meant to use: “Some participants had the feeling of shame about going back to their families. They felt it would be intimidating to their family, as well as to themselves, to go back home after having had homeless life.” Possibly intimidating is the expression for what the participants feel going home in terms of how their family would receive them and treat them, but is their presence actually intimidating to their families? If so, how? Might it be embarrassing to the families to have the woman home rather intimidating?

Discussion: First paragraph mentions cultural peculiarities – are they peculiarities actually? Every location has its own cultural norms – so what makes the study unique is not the cultural peculiarities regarding gender norms – but the study analysis is situated within the context of gender norms in Ethiopia. Or is it that with this sentence the authors are referring specifically to child marriage? Best to make this clear.

Sentence: “However, the data were presented as were described by participants without much interpretation.” This is not generally considered a good thing, as interpretation is the final step of qualitative analysis and therefore essential. I think that the authors did do interpretation as they categorized the factors affecting homelessness. Best to just remove this sentence. Another way to deal with the fact that some authors may have been predisposed to certain thoughts about the data is that the study and paper includes a number of authors, and discussion among the authors of the data as data were analyzed and the paper constructed would/could have mitigated bias of one or two authors.

This is a paper about an important topic, using rich qualitative data. Overall the paper reads much more smoothly than the original version read. However, to ensure that the authors’ arguments are expressed as clearly as possible for publication, there are still some syntax edits needed. Check the entire manuscript for tense to keep tenses consistent in past or present – at times both are used in the same sentence. Check sentences for missing words such as ‘a’ and ‘an’ and ‘the’ and ‘at’ and necessary ‘s’ for pluralization in some places. Note that in some sentences ‘was’ is used where the plural ‘were’ is needed. Typically contractions (e.g. couldn’t) are not used so best to check the paper for contractions and use the two words instead. Possibly the journal has an editing service to help with these issues.

Reviewer #2: The reason for not all underlying data are made available is explained by the authors, and their explanation is fully acceptable.

7. PLOS authors have the option to publish the peer review history of their article (what does this mean?). If published, this will include your full peer review and any attached files.

Reviewer #1: No

Reviewer #2: **Yes: **Gunnar Aksel Bjune

---

## [Author Response · Author response to Decision Letter 1]

18 Aug 2020

Response to reviewers

Please find herein the point-by-point responses to your comments:

1. Sampling: The word ‘case’ was inserted after ‘typical’ in the sampling strategy subsection as you commented. Check revised manuscript with track changes. About inserting a reference to help justify the method, we think this would not be very vital to this manuscript as our intention is not to teach about sampling methods in qualitative studies. Citations were inserted to provide credit to authorities whose ideas and strong arguments we used to justify our study. Providing a reference as you suggested, we believe, is not relevant here. 

2. Setting: In the ‘setting’ sub-section paragraph 2 the statement describing about the shelter being for women and elderly men the following amendment has been made. ‘The shelter was established to provide services to homeless adult women and their dependent children if they had any, and elderly men’. The fact is the shelter took in homeless women together with their children and provided basic services to the children, as well as made it possible to attend school at KG and above.

3. Ethics approval: The unnecessary detailed description about the contents of the participant information sheet was deleted from the subsection ‘ethics approval and consent to participate’ as you suggested. Check revised manuscript with track changes.

4. Results: The following points were addressed:

• The spelling errors in the first paragraph are corrected. ‘on of the participants—‘ was mad ‘one of the ---,‘ ‘---- at least one children—‘ was changed to ‘---at least one child---,‘ ‘step-parent’ was consistently used. Please check manuscript with track changes.

• In the ‘predisposing factors’ sub-section the paragraph which started by the statement ‘some participants had experienced exploitative work conditions---‘ the second statement has been edited to make it better. Please check manuscript with track changes.

• The sentence ‘In some cases participants were opted---‘ has been replaced by ‘In some cases participants opted ---‘ as you commented. Shown by track changes.

• In the paragraph which starts by ‘Some participants had the feeling of shame----‘ the second sentence was changed to ‘They felt it would be an embarrassment to their family, as well as to themselves,----‘ (Shown by track changes)

• The term ‘respondent’ has been replaced by ‘participant’ in several paragraphs of the ‘results’ section. Please check manuscript with track changes.

• Missed words were inserted and grammatical corrections including tense, non-use of contraction and the correct use of plural were made in several parts of the ‘results’ section as shown by track changes.

5. Discussion: The following amendments were made:

• In the first paragraph of the ‘discussion’ section the second sentence was changed into ‘the study was conducted in a country which had cultural norms which gave women a lower and submissive status and child marriage was widely practiced,’ in order to avoid ambiguities. 

• The sentence which started by ‘However, the data were presented as were----‘ was removed and replaced by the sentence ‘However, as much participant narrations as possible were provided to give the reader better understanding’. Please check track changes.

---

## [Editor Report · Decision Letter 2]

20 Aug 2020

Pathways through homelessness among women in Addis Ababa, Ethiopia: A qualitative study

PONE-D-20-15059R2

Dear Dr. Haile,

We’re pleased to inform you that your manuscript has been judged scientifically suitable for publication and will be formally accepted for publication once it meets all outstanding technical requirements.

Kind regards,

Mellissa H Withers, PhD, MHS

Academic Editor

PLOS ONE
---

## [Editor Report · Acceptance letter]

21 Aug 2020

PONE-D-20-15059R2 

Pathways through homelessness among women in Addis Ababa, Ethiopia: A qualitative study 

Dear Dr. Haile:

I'm pleased to inform you that your manuscript has been deemed suitable for publication in PLOS ONE. Congratulations! Your manuscript is now with our production department. 

Kind regards, 

on behalf of

Dr. Mellissa H Withers 

Academic Editor

PLOS ONE